

# Wireless sensor localization based on distance optimization and assistance by mobile anchor nodes: a novel algorithm

Hui Yang

School of Media Engineering, Lanzhou University of Arts and Science, Lanzhou, Gansu, China

## ABSTRACT

Wireless sensor networks (WSNs) have wide applications in healthcare, environmental monitoring, and target tracking, relying on sensor nodes that are joined cooperatively. The research investigates localization algorithms for both target and node in WSNs to enhance accuracy. An innovative localization algorithm characterized as an asynchronous time-of-arrival (TOA) target is proposed by implementing a differential evolution algorithm. Unlike available approaches, the proposed algorithm employs the least squares criterion to represent signal-sending time as a function of the target position. The target node's coordinates are estimated by utilizing a differential evolution algorithm with reverse learning and adaptive redirection. A hybrid received signal strength (RSS)-TOA target localization algorithm is introduced, addressing the challenge of unknown transmission parameters. This algorithm simultaneously estimates transmitted power, path loss index, and target position by employing the RSS and TOA measurements. These proposed algorithms improve the accuracy and efficiency of wireless sensor localization, boosting performance in various WSN applications.

## INTRODUCTION

Wireless sensor networks (WSNs) have become an increasingly substantial technology with pivotal application potential and commercial value (*Navarro et al., 2024*; *Cong, Thi & Thanh, 2024*; *Dubey et al., 2023*). This can be attributed to advancements in microelectronics, sensing technology, and embedded technology, which have greatly enhanced the performance of sensor devices. As a result, research institutions and technology enterprises worldwide have shown increased interest in WSNs. A solid outcome of this growing interest is the development of TinyOS, an open-source sensor network operating system, by the University of California, Berkeley (*Marah, Kardas & Challenger, 2021*; *Amjad, 2016*).

In particular, localization is a pivotal research area in WSNs, especially for location-based services (*Gebremariam, Pand & Indu, 2023*; *Sun et al., 2023*). Researchers aim to improve localization algorithms for both target and node localizations (*Gebremariam, Panda & Indu, 2022*). Target localization involves determining the positions of unknown

Corresponding author
Hui Yang, yanghuiyyh@163.com

target nodes by measuring specific signal features (*Zhang et al., 2022*). The anchor node captures information from source devices and sends the measured data to the data fusion center to predict the target node's position. Various measurement techniques utilize signal characteristics to predict target positions, such as time-of-arrival (TOA), time difference of arrival (TDOA), and angle of arrival (AOA). The TOA and TDOA-based algorithms offer higher accuracy and save cost than AOA-based algorithms but require time synchronization. The current research on the TOA-based algorithms aims to overcome the challenges of time synchronization. Linearization-based weighted least squares (WLS) algorithms, convex optimization methods employing Semidefinite Programming (SDP) relaxation, and proximal alternating minimization positioning (PAMP) algorithms are suggested to address these synchronization issues (*Wan & Lu, 2022*).

Researchers have explored location algorithms based on hybrid measurements, which combine multiple measurement techniques to enhance positioning accuracy. Hybrid algorithms have been researched by combining the TOA and AOA, the RSS and AOA, or the RSS and round trip time (RTT) measurements, respectively (*Zefu, Yiwen & Wenge, 2023*). These algorithms employ robust least squares multilateral localization (RLSM), relaxed likelihood function, and iterative generalized trust region sub problem (GTRS) to predict target node positions. They also address challenges such as Non-Line-of-Sight (NLOS) errors and unknown transmission parameters.

Range-based localization algorithms find the coordinates of unknown nodes in WSNs by measuring separations or positions between devices (*Luomala & Hakala, 2022*; *Wye et al., 2021*). On the other hand, non-ranging-based localization algorithms predict node positions without directly gauging distances or angles. The DV Hop algorithm, known for its simplicity, has progressed in various aspects, including the number of hops, average hop distance, and coordinate calculation methods. Metaheuristic algorithms, such as particle swarm optimization (PSO), differential evolution (DE), and cuckoo search (CS), have been utilized to compute coordinates.

The research aims to enhance localization accuracy in WSNs through innovative algorithms. An asynchronous TOA target localization algorithm based on differential evolution is proposed. Unlike available approaches, it models signal-sending time as a function of target position by employing the least squares method. The target node's coordinates are estimated by implementing a differential evolution algorithm with reverse learning and adaptive redirection. Additionally, a hybrid RSSTOA algorithm addresses the challenge of unknown transmission parameters by simultaneously estimating power, path loss index, and target position. These algorithms enhance wireless sensor localization accuracy and efficiency in various WSN applications.

The research contributes to the literature by adding an asynchronous TOA target and employing the least squares criterion to represent signal-sending time as a function of the target position. Then, the target node's coordinates are estimated by utilizing a differential evolution algorithm with reverse learning and adaptive redirection. Thus, the challenge of

unknown transmission parameters is addressed by a hybrid received signal strength (RSS)-TOA target localization algorithm.

The rest of the article is structured as follows: "Asynchronous TOA-based Target Location Algorithm" presents the background of the asynchronous TOA target localization algorithm. "Hybrid RSS-TOA-based Target Location Algorithm with Unknown Transmission Parameters" is allocated to a hybrid RSSTOA algorithm that addresses the challenge of unknown transmission parameters by simultaneously estimating power, path loss index, and target position. The research is concluded in "Conclusion".

## ASYNCHRONOUS TOA-BASED TARGET LOCATION ALGORITHM

In TOA-based localization algorithms, anchor nodes need to determine the exact timing of signal transmission from the target node to find the propagation time and compute the distance between the anchor nodes and the target node. However, this requirement for strict time synchronization among main and destination nodes causes complexity and deployment costs to the network. Therefore, the problem of target localization based on asynchronous TOA needs further investigation, where synchronization is only required among the anchor nodes, not between the anchor nodes and the target node. In other words, the anchor nodes do not need to determine the timing of transmitting the signal from the destination node. Currently, two main approaches exist to address the target localization problem based on the asynchronous TOA. One approach is to subtract pairwise TOA measurements to disregard the unfamiliar timing of signal transmission and transform the problem into a TDOA localization problem. However, this transformation introduces an increase of 3 dB in measurement noise. The alternative approach is to collectively calculate the timing of signal transmissions and the destination position by utilizing the TOA assessments and constructing anchor node coordinates. Studies have indicated that both TDOA and TOA-dependent target localization predicaments involve extremely nonlinear and non-convex objective functions. To handle such nonlinearity and avoid iterative search algorithms not getting trapped in local optima, it is common to relax the non-convex optimization problem into a convex optimization one. However, this relaxation may result in performance loss and introduce unreasonable computational burdens. In contrast to conventional algorithms, the suggested asynchronous TOA target localization algorithm based on differential evolution does not require easing the non-convex localization dilemma into a convex predicament. Instead, it implements differential evolution with reverse learning and adaptive redirection to resolve for the target node's coordinates.

A 2D wireless sensor network is composed of N anchor nodes and a target node. The anchor nodes are defined as $a_i = [a_{i1}, a_{i2}]^T$, where $i = 1, \ldots, N$. The coordinates of the target node are denoted as $s = [s_1, s_2]^T$, which are unknown. The target node emits radio signals, and each anchor node extracts TOA measurements from the received signals. The

asynchronous TOA measurement score at the i[th] anchor node can be modeled by employing Eq. (1). The maximum likelihood estimation of the signal transmission time t0 and the target node coordinates s are presented in Eq. (2).

$$t_i = t_0 + \frac{1}{c} \| s - a_i \| + \epsilon_i, i = 1, \ldots, N \tag{1}$$

$$(\widehat{s}, \widehat{t_0}) = min_{s,t_0} \sum_{i=1}^{N} \frac{\left(t_i - t_0 - \frac{1}{c} \| s - a_i \|\right)^2}{\sigma_{t_i}^2} \tag{2}$$

The non-convex TOA localization problem is resolved by employing the differential evolution algorithm. A target function needs to be constructed for optimization to apply the differential evolution algorithm. Constructing the target function based on Eqs. (3) and (4) yield Eq. (5), which delineates the localization problem, as shown in Eq. (6). Then, the coordinates of the target node can be determined by employing the differential evolution algorithm with reverse learning and adaptive reorientation.

$$\widehat{t_0} = \frac{1}{N} \sum_{i=1}^{N} \left( t_i - \frac{1}{c} \| s - a_i \| \right) \tag{3}$$

$$\sum_{i=1}^{N} \left( t_i - \frac{1}{N}\sum_{j=1}^{N}\left( t_j - \frac{1}{c} \| s - a_j \| \right) - \frac{1}{c} \| s - a_i \| \right)^2 \tag{4}$$

$$f(s) = \sum_{i=1}^{N} \frac{\left( t_i - \frac{1}{N}\sum_{j=1}^{N}\left( t_j - \frac{1}{c} \|s - a_j\| \right) - \frac{1}{c} \|s - a_i\| \right)^2}{\sigma_{t_i}^2} \tag{5}$$

$$\widehat{s} = min_s f(s) \tag{6}$$

The distinct growth mechanism is a random exploration mechanism impacted by the developmental hypothesis, initially proposed by *Storn & Price (1997)*. It has discovered extensive implementations in engineering advancement domains such as information processing, asset arrangement, manufacturing blueprint, imitation neural networks, and energy structure advancement. This mechanism imitates the evolution of communities in nature through mutation, interbreeding, and assortment procedures. Reverse learning is a search strategy based on the concept of oppositional computation. Inspired by the antagonistic relationship between objects, reverse learning aims to accelerate convergence to the optimal solution by simultaneously considering the original and its opposite to find better candidate solutions (*Choi, Togelius & Cheong, 2021*).

The steps of the differential evolution algorithm with reverse learning and adaptive reorientation are presented as follows:

(1) Initial population generation: an L number of candidate individuals denoted by X with dimension D are generated by employing Eq. (7) where d = 1, 2, …, D, and since a 2D wireless sensor network is considered so, D = 2 and rand represents a random variable that produces numbers uniformly distributed in the range [0, 1], *i.e.*, rand ~U (0, 1), ubd and lbd denote the upper and lower bounds of the d-th dimension of the target node

coordinates, respectively. After generating an L number of candidate individuals, reverse individuals are constructed by utilizing the reverse learning strategy, as shown in Eq. (8). Then, the population evolves iteratively for G different generations through mutation, crossover, adaptive reorientation, and selection operations.

$$\left\{ X_l^- = \left[ x_{l,1}^-, x_{l,2}^-, \ldots, x_{l,D}^- \right] \right\}_{l=1}^L \tag{7}$$

$$x_{l,d}^- = lb_d + ub_d - x_{l,d}^+ \tag{8}$$

(2) Mutation and crossover: The mutation is represented by Eq. (9) where $F \in [0, 1]$ represents the scaling factor, g represents the g-th iteration, and o, p, q $\in \{1, 2, \ldots, L\}$ denote randomly selected indices, with the condition $0 \neq p \neq q$. The crossover operation is defined as Eq. (10).

$$V_l^g = X_o^g + F\left( X_p^g - X_q^g \right) \tag{9}$$

$$u_{l,d}^g = \begin{cases} v_{l,d}^g & \text{if } rand \leq CR \\ x_{l,d}^g & \text{otherwise} \end{cases} \tag{10}$$

(3) Adaptive reorientation and selection: Considering the information contained in the current population, the reorientation process ensures that individuals do not exceed the upper and lower bounds of the target node coordinates. It is generated by utilizing Eq. (11). Among the number of 2L individuals, only L individuals with the smallest fitness scores are picked as the population for the (g+1)-th iteration, as shown in Eq. (12).

$$\breve{u}_{l,d}^g = \begin{cases} u_{l,d}^g & \text{if } lb_d \leq u_{l,d}^g \leq \& \leq \boldsymbol{ub}_d \\ rand\left( ub_d^g - lb_d^g \right) + lb_d^g & \text{otherwise} \end{cases} \tag{11}$$

$$X_l^{g+1} = \min_{X_l^g, \breve{U}_l^g} \left\{ f\left( X_l^g \right), f\left( \breve{U}_l^g \right) \right\} \tag{12}$$

(4) Selection of the best individual for coordinate estimation: After G iterations, the individual with the smallest fitness score (*i.e.*, the best individual) is picked from the population as the predicted coordinates of the target node, as shown in Eq. (13).

$$\widehat{X} = \min_{X_l^G} \left\{ f\left( X_l^G \right) \right\}_{l=1}^L \tag{13}$$

In the proposed localization algorithm (DEOR), simulations were conducted in four different scenarios by running MATLAB R2020a. The algorithms proposed in previous studies were analyzed and compared in the article (*Xu, Ding & Dasgupta, 2011*; *Chen, Yao & Peng, 2020*). The 2LS algorithm was implemented by running the CVX toolbox with the SeDuMi solver.

Scenario 1: Eight anchor nodes were deployed with the following coordinates: $a_1 = [400, 0]^T$, $a_2 = [400, -400]^T$, $a_3 = [-400, 400]^T$, $a_4 = [-400, -400]^T$, $a_5 = [800, 800]^T$, $a_6 = [800, -800]^T$, $a_7 = [-800, 800]^T$, $a_8 = [-800, -800]^T$ (unit: m). The target node was located at [30, 10] within the convex hull formed by the anchor nodes. Each algorithm's root mean square

| Table 1 Average calculation time of each algorithm in Scenario 1 ($1/\sigma^2 = 10$ dB). | |
| --- | --- |
| **Algorithm** | **Average calculation time ($10^{-2}$ s)** |
| 2LS | 52 |
| PAMP | 0.9 |
| DEOR(50) | 1.3 |
| DEOR(100) | 2.5 |

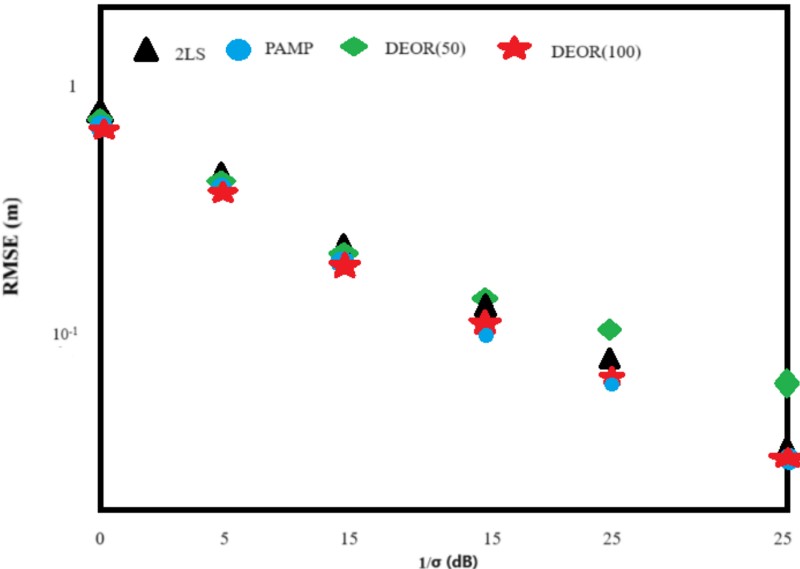

**Figure 1 RMS error of each algorithm in Scenario 1.**

error (RMSE) reduces as the measured noise variance decreases in Fig. 1. Due to the relatively ideal conditions in Scenario 1, all algorithms exhibit high positioning accuracy, with small differences in RMSE. In Table 1, when compared to the 2LS algorithm, the DEOR algorithm and the PAMP algorithm have advantages regarding the computational time, with the PAMP algorithm having the shortest computation time. Furthermore, the DEOR algorithm needs more iterations to determine the optimal score, resulting in an increased computational time when the scenario is relatively ideal and the measured noise is low. Both Fig. 1 and Table 1 present the outcome when DEOR(50) and DEOR(100) are compared. This also reflects the close relationship between the computational cost of the DEOR algorithm and the number of iterations.

Scenario 2: The deployment of anchor nodes remained the same as in Scenario 1. The target node was randomly generated within the monitoring area $[-1,200, 1,200] \times [-1,200, 1,200]$. Based on the combination of Figs. 1 and 2, the RMSE of all algorithms increases in Scenario 2 when compared to the relatively ideal Scenario 1, where the target nodes are randomly deployed. On the other hand, unlike Figs. 1 and 2 shows that the DEOR algorithm and the PAMP algorithm have significantly higher positioning accuracy than the 2LS algorithm. Table 2 shows that both the DEOR algorithm and the PAMP algorithm

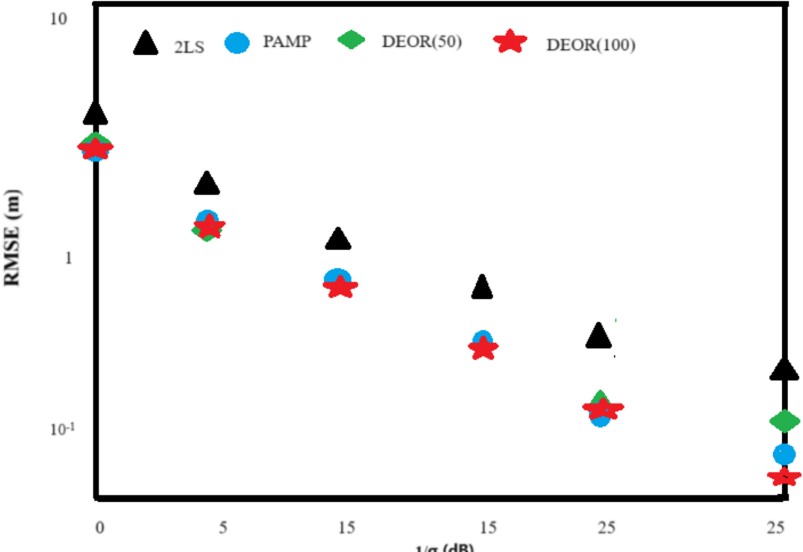

**Figure 2 RMS error of each algorithm in Scenario 2.**

**Table 2 Average calculation time of each algorithm in Scenario 2 ($1/\sigma^2 = 10$ dB).**

| Algorithm | Average calculation time ($10^{-2}$ s) |
|---|---|
| 2LS | 59 |
| PAMP | 1.6 |
| DEOR(50) | 1.2 |
| DEOR(100) | 2.3 |

require less computation time when compared to the 2LS algorithm. By comparing Tables 1 and 2 for Scenario 2, the computation time required by the PAMP algorithm increases significantly, no longer providing a clear advantage over the DEOR algorithm. This also reflects that the computational time of the DEOR algorithm is independent of the node deployment.

Scenario 3: The deployment of anchor nodes remained the same as in Scenario 1. The monitoring area was set to $[1,200, 3,200] \times [-800, 800]$, and the target node was located at $[3,000, 10]^T$, outside the convex hull formed by the anchor nodes. Figure 3 shows that the positioning accuracy of all algorithms decreases when compared to Scenario 1, as the target node is located outside the convex hull formed by the anchor nodes due to its larger distance. Among the algorithms, the DEOR algorithm exhibits the smallest RMSE. Particularly, the PAMP algorithm, which performs exceptionally well in Scenarios 1 and 2, shows the poorest positioning accuracy in this case. This mainly occurs because the PAMP algorithm consistently converges to local minima in Scenario 3. According to Table 3, the computational time of the DEOR algorithm is shorter than that of the PAMP algorithm and the 2LS algorithm, respectively. This confirms that the computation time of the DEOR algorithm is independent of the node deployment.

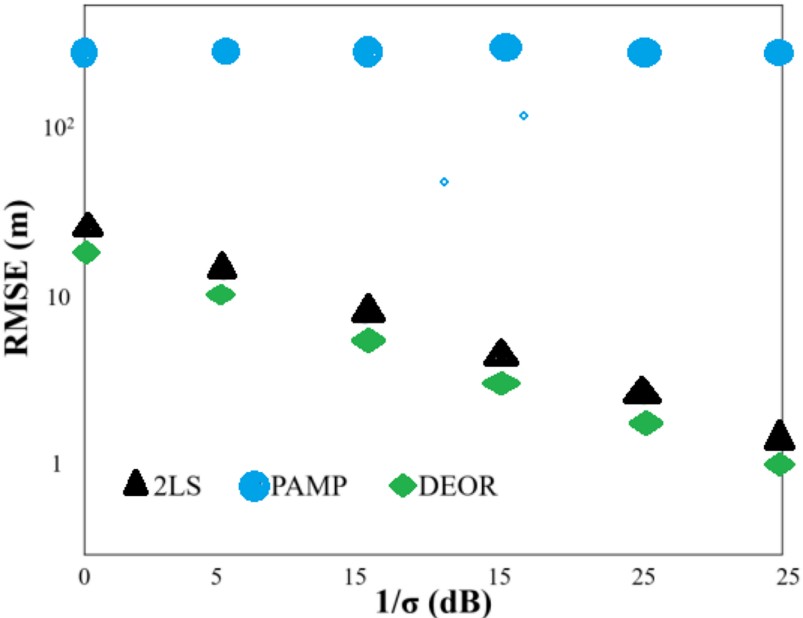

**Figure 3 RMS error of each algorithm in Scenario 3.**

**Table 3 Average calculation time of each algorithm in Scenario 3 ($1/\sigma^2 = 10$ dB).**

| Algorithm | Average calculation time ($10^{-2}$ s) |
|---|---|
| 2LS | 60 |
| PAMP | 5.5 |
| DEOR | 1.2 |

Scenario 4: The anchor and target nodes were randomly deployed within the monitoring area [800, 800] × [−800, 800]. Figure 4 illustrates that the DEOR algorithm has a smaller RMSE when compared to the 2LS algorithm and the PAMP algorithm, respectively. Due to the random deployment of anchor nodes and target nodes within the monitoring area, the PAMP algorithm obtains a large number of local minima points instead of the global minima point in multiple experiments. Table 4 indicates that the DEOR algorithm requires the least computation time among the other three algorithms.

The four sets of experiments conducted in the four scenarios show that the DEOR algorithm exhibits high positioning accuracy, low computation time, and independence from node deployment in each scenario, demonstrating robustness. When compared to the 2LS algorithm that utilizes convex optimization techniques, the DEOR algorithm achieves higher positioning accuracy and requires less computation time. Although in Scenarios 1 and 2, the positioning accuracy of the PAMP algorithm is similar to that of the DEOR algorithm, and in Scenario 1, the computation time of the PAMP algorithm is significantly lower than that of the DEOR algorithm. However, in Scenarios 3 and 4, the DEOR algorithm outperforms the PAMP algorithm in terms of positioning accuracy, and the PAMP algorithm no longer has a computational advantage at this point.

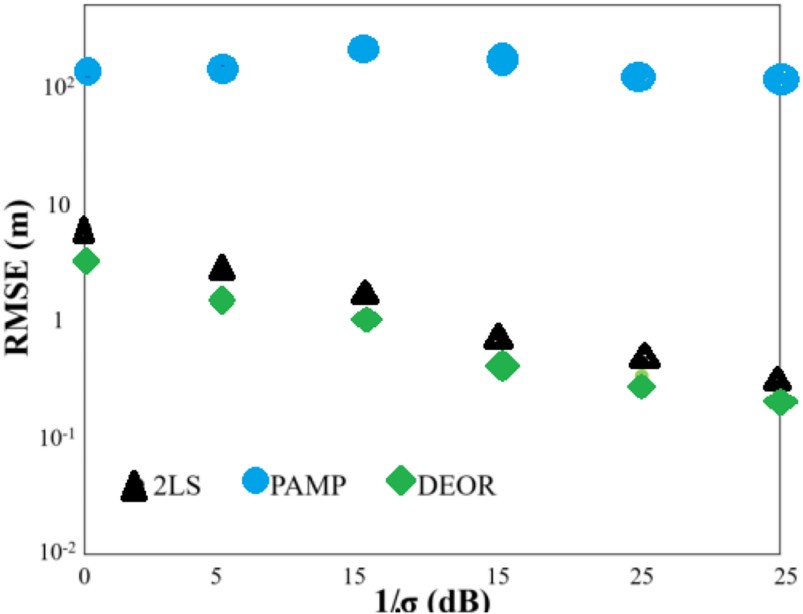

**Figure 4 RMS error of each algorithm in Scenario 4.**

**Table 4 Average calculation time of each algorithm in Scenario 4 ($1/\sigma^2$ = 10 dB).**

| Algorithm | Average calculation time ($10^{-2}$ s) |
|---|---|
| 2LS | 60 |
| PAMP | 5.5 |
| DEOR | 1.2 |

# HYBRID RSS-TOA-BASED TARGET LOCATION ALGORITHM WITH UNKNOWN TRANSMISSION PARAMETERS

In recent years, the fusion of two types of measurements in hybrid localization systems has attracted widespread attention. These hybrid localization systems aim to advance nodes' placement precision by utilizing every measurement approach's abilities. RSS is more suitable for short-range measurements, while TOA is more suitable for long-range measurements (*Coluccia & Fascista, 2018*). Therefore, based on the proposed asynchronous TOA target localization algorithm employing differential evolution, this section introduces RSS measurements and proposes a hybrid RSS-TOA target localization algorithm based on differential evolution, fully utilizing both measurement techniques' advantages. Unlike most studies that assume precise prior knowledge of transmission power and path loss exponent, the proposed algorithm addresses the localization problem under a more realistic assumption that the exact scores of transmission power and path loss exponent are unknown. For the same non-convex and nonlinear hybrid RSS-TOA localization problem, the differential evolution algorithm with reverse learning and adaptive redirection is implemented to jointly predict the target position and the RSS transmission parameters.

When a wireless sensor network composed of N anchor nodes and one target node is considered, the anchor nodes are defined as $a_i = [a_{i1}, a_{i2}]^T$, and $s_i = [s_1, s_2]^T$ represents the coordinates of the target node. The target node transmits a wireless signal, and each anchor node extracts the received signal strength (RSS) measurements from the received signal. According to the log-normal shadowing model, the power received by the i-th anchor node, $p_i$, is given by Eq. (14), where $p_0$ represents the transmission capability of the aimed node measured at distance $d_0$, which is usually taken as $d_0 = 1$ m. When the transmission power of the target node is unknown, predicting the transmission power is equivalent to estimating $p_0$. $\gamma$ represents the path loss exponent, and $||s−a_i||$ denotes the Euclidean distance between the target node and the i-th anchor node. $\varepsilon_i$ represents the measurement noise of the RSS and accounts for the log-normal shadowing effect, which is modeled as a zero-mean Gaussian random variable, $\varepsilon \sim N(0,\sigma_{ri2})$, where $\sigma_{ri}$ represents the standard deviation of the RSS measured noise. Based on Eq. (14), the maximum likelihood estimation for the RSS-based localization problem is expressed by Eq. (15).

$$p_i = p_0 - 10 \, \gamma \, \log_{10} \frac{||s - a_i||}{d_0} + \varepsilon_i, i = 1, \ldots, N \tag{14}$$

$$(\hat{s}, \hat{p}_0, \hat{\gamma}) = \min_{s, p_0, \gamma} \sum_{i=1}^{N} \frac{\left(p_i - p_0 + 10 \, \gamma \, \log_{10}||s - a_i||\right)^2}{\sigma_r^2} \tag{15}$$

All measurement scores of the RSS and the TOA are represented by $\Psi = [t_1 \ldots, t_n, P_1, \ldots, P_N]$, and the parameters to be estimated are denoted by $\xi = [s^T, p_0, \gamma, t_0]$. Assuming that the errors in the RSS and the TOA measurements are independent, then the joint probability density function of all measurement scores represented by Eq. (16). Research has shown that the RSS and TOA measurements extracted from the same signal exhibit weak correlations, which can be combined in multiple ways. Therefore, the assumption of uncorrelated measurements does not compromise generality (*Tomic & Beko, 2019*). From Eq. (16), the maximum likelihood estimation of $\xi$, denoted as $\xi$ in Eq. (17), can be derived.

$$\mathcal{P}(\psi \mid \xi) = \prod_{i=1}^{N} \frac{1}{\sqrt{2\pi}\sigma_{r_i}} \exp\left\{ \frac{-\left(p_i - p_0 + 10 \, \gamma \, \log_{10}||s - a_i||\right)^2}{2\sigma_{r_i}^2} \right\} \tag{16}$$

$$\hat{\xi} = \min_{\xi} \sum_{i=1}^{N} \frac{(ct_i - ct_0 - ||s - a_i||)^2}{\sigma_{dis_i}^2} + \sum_{i=1}^{N} \frac{\left(p_i - p_0 + 10 \, \gamma \, \log_{10}||s - a_i||\right)^2}{\sigma_{r_i}^2} \tag{17}$$

Then, based on whether the transmission power and path loss exponent are known, the target localization problem can be categorized into three cases: (1) unknown transmission power, known path loss exponent; (2) known transmission power, unknown path loss exponent; (3) unknown transmission power and path loss exponent. The localization problem in the three circumstances can be unifiedly represented by Eqs. (18) and (19). Finally, as mentioned earlier, the target node's coordinates and unknown transmission parameters are resolved by employing the differential evolution algorithm with reverse learning and adaptive redirection. However, in the case of unknown transmission parameters, the target localization algorithm based on a hybrid RSS-TOA approach also

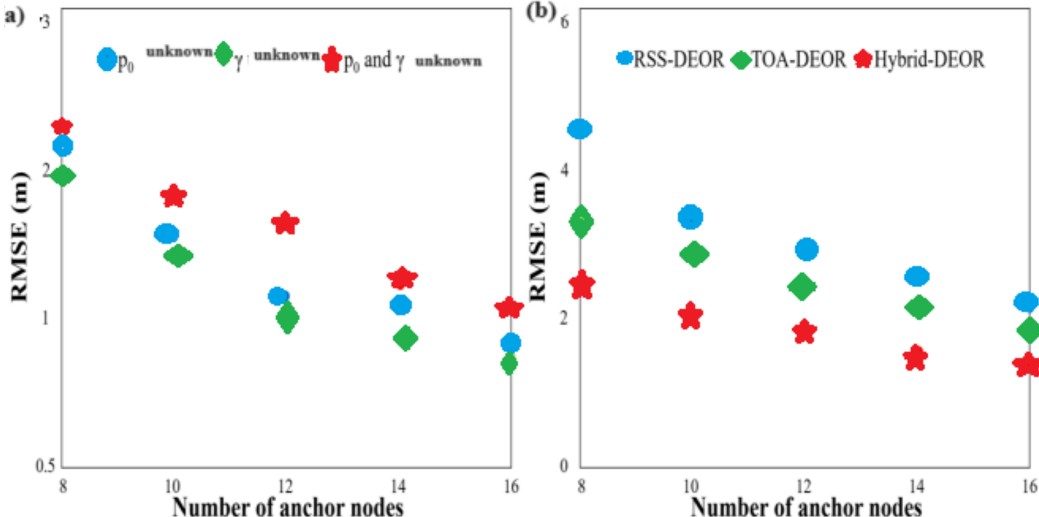

**Figure 5** RMSE of (A) hybrid-DEOR algorithm in three cases and (B) a different number of anchor nodes.                                              

requires estimating the transmission parameters for the RSS. Therefore, the problem dimension for cases (1) and (2) equals 3, 4 while for case (3).

$$\hat{\xi}_k = min_{\xi_k} f(\xi_k), k = \{1, 2, 3\} \tag{18}$$

$$f(\xi_k) = \sum_{i=1}^{N} \frac{\left(ct_i - c\frac{1}{N}\sum_{j=1}^{N}\left(t_j - \frac{1}{c}||\boldsymbol{s} - \boldsymbol{a}_j||\right) - ||\boldsymbol{s} - \boldsymbol{a}_i||\right)}{\sigma_{dis_i}^2} \\ + \sum_{i=1}^{N} \frac{(p_i - p_0 + 10\,\gamma\,log_{10}||\boldsymbol{s} - \boldsymbol{a}_i||)^2}{\sigma_{r_i}^2} \tag{19}$$

The suggested target localization algorithm with unknown transmission parameters based on the differential evolution algorithm is referred to as Hybrid-DEOR. The target localization algorithm with unknown RSS transmission parameters proposed in *Caceres Najarro et al. (2020)* is called RSS-DEOR. The target localization algorithm based on the differential evolution algorithm for asynchronous TOA is called TOA-DEOR. Simulation experiments are carried out to assess the effectiveness of three localization approaches.

First, the effectiveness of the Hybrid-DEOR algorithm is verified in all three cases through simulations. The number of anchor nodes is set to eight through six with a step size of 2, and the standard deviation of measurement noise is set to 2. The comparison of RMSE for the Hybrid-DEOR algorithm in all three cases is given in Fig. 5A. The RMSE decreases since the quantity of anchor nodes escalates. The RMSE is lower when only one transmission parameter is unknown when compared to the case where both transmission parameters are unknown, and the RMSE is lower when the path loss exponent is unknown when compared to the case where the transmission power is unknown. The suggested Hybrid-DEOR algorithm is effective in all three cases. Unless otherwise specified, the following RSS-DEOR and Hybrid-DEOR algorithms are simulated under unknown

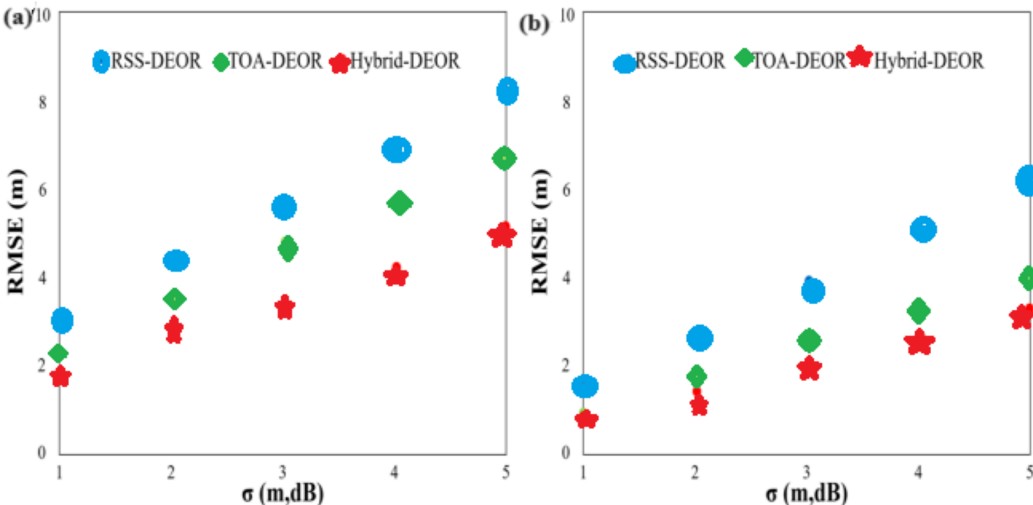

**Figure 6 RMSE of different measurement noise standard deviations: anchor nodes are deployed (A) randomly and (B) evenly.**

transmission power and path loss exponent conditions. Figure 5B compares the RMSE scores for the RSS-DEOR, the TOA-DEOR, and the Hybrid-DEOR algorithms with different numbers of anchor nodes. The RMSE scores of the three algorithms decrease since the quantity of anchor nodes escalates. Increasing the anchor nodes' number helps the DEOR algorithm search for the optimal solution.

On the other hand, the Hybrid-DEOR algorithm achieves higher localization accuracy when compared to the RSS-DEOR and the TOA-DEOR algorithms under different numbers of anchor nodes. For example, when the number of anchor nodes is 8, the RMSE scores of the RSS-DEOR, the TOA-DEOR, and the Hybrid-DEOR algorithms are approximately 4.531, 3.262, and 2.402 m, respectively. The Hybrid-DEOR approach enhances the positioning precision by approximately 46.99% and 26.36% when compared to the other two algorithms. Since all three algorithms employ the DEOR, the Hybrid-DEOR algorithm achieves higher localization accuracy by fully utilizing the advantages of the RSS in short-distance measurements and the TOA in long-distance measurements.

The number of anchor nodes is set to 8, and the standard deviation of measured noise ranges from 1 to 5 with a step size of 1. Two different anchor node deployment scenarios are considered: random deployment in a 40 m × 40 m area and uniform deployment. The coordinates of anchor nodes are set to $(0, 0)^T$, $(0, 40)^T$, $(40, 40)^T$, $(40, 0)^T$, $(10, 10)^T$, $(10, 30)^T$, $(30, 10)^T$, and, $(30, 30)^T$, respectively. Figure 6 compares the RMSE scores for all algorithms under different standard deviations of measured noise. Figure 6A corresponds to random anchor node deployment, while Fig. 6B corresponds to uniform anchor node deployment. The RMSE scores of all approaches increase as the standard deviation of measured noise increases. The Hybrid-DEOR algorithm achieves higher localization accuracy than those of the RSS-DEOR and TOA-DEOR algorithms under different standard deviations of measured noise. For example, when the standard deviation of measured noise is three with random anchor node deployment, the RMSE scores of the

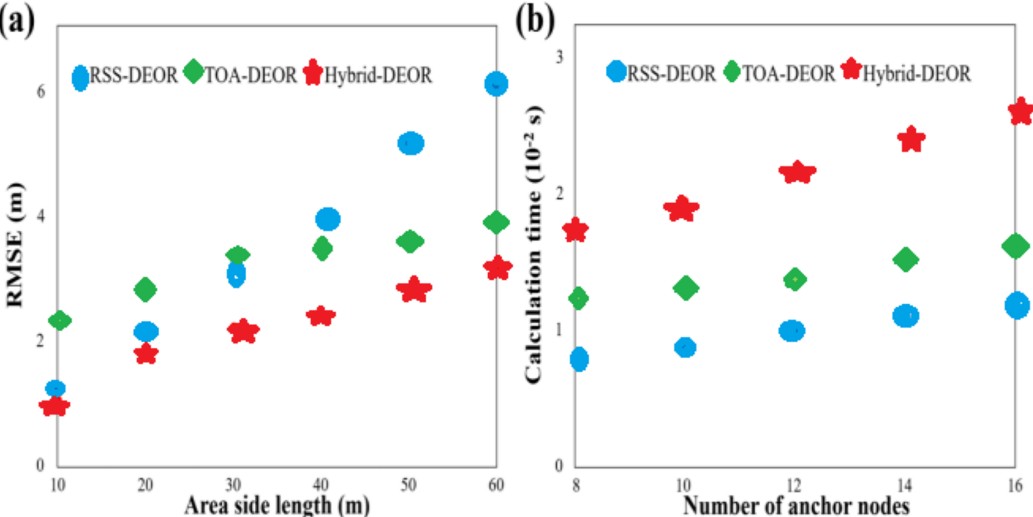

**Figure 7** (A) RMSE of different area side lengths. (B) Calculation time under different numbers of anchor nodes.

RSS-DEOR, the TOA-DEOR, and the Hybrid-DEOR algorithms are approximately 5.558, 4.649, and 3.329 m, respectively. The Hybrid-DEOR algorithm improves the localization accuracy by approximately 40.10% and 28.39% when compared to the other two algorithms. By comparing Figs. 6A and 6B, the RMSE scores decrease when anchor nodes are uniformly deployed when compared to random deployment. This indicates that changing the deployment of anchor nodes can effectively improve the positioning precision of the algorithms. For example, when the standard deviation of measured noise is three with uniform anchor node deployment, the RMSE score of the Hybrid-DEOR algorithm is 2.066 m, approximately 37.94% higher than that of random anchor node deployment.

The number of anchor nodes is set to 8, the standard deviation of measured noise is 2, and the range of the width of the region is constructed from 10 to 60 m with a step size of 10 m. Figure 7A compares the RMSE scores of the RSS-DEOR, TOA-DEOR, and Hybrid-DEOR, respectively algorithms under different side lengths of the area. As the side length of the area increases, resulting in a larger area and sparser distribution of nodes, the localization accuracy decreases. However, in all cases, the Hybrid-DEOR algorithm consistently achieves the smallest RMSE when compared to the other two algorithms. Additionally, there is a boundary at about 30 m. When the side length is smaller than 30 m, the localization accuracy of the RSS-DEOR is higher than that of the TOA-DEOR, when the side length is larger than 30 m, the localization accuracy of the RSS-DEOR is lower than that of the TOA-DEOR. This also indicates that the TOA is more suitable for long-distance measurements, while the RSS is more suitable for short-distance measurements. The number of anchor nodes is set to 8–16 with a step size of 2, and the standard deviation of measured noise is set to 2. Figure 7B compares computational time for all algorithms under different numbers of anchor nodes. The computational time increases as the number of anchor nodes increases. Apart from the iteration number and population size,

anchor nodes' quantity is another important factor impacting the computational time because the calculation of individual fitness scores involves accumulated information from all anchor nodes. On the other hand, the Hybrid-DEOR algorithm requires the highest computational time since it utilizes the RSS and the TOA information. Due to the inclusion of the estimation of signal transmission time in the objective function of the TOA-DEOR algorithm, it requires more computational time than the RSS-DEOR.

# CONCLUSION

Overall, the research presents a wireless sensor localization algorithm that addresses wireless sensor networks' target and node localization challenges. It provides a comprehensive analysis of range-based and non-range-based location algorithms, laying the foundation for further research. The study primarily focuses on the target location problem by employing asynchronous TOA. An innovative asynchronous TOA target location algorithm is suggested, leveraging the differential evolution algorithm to find the target node's coordinates effectively. Simulation results demonstrate the algorithm's superior positioning accuracy and lowered computational time when compared to available algorithms. Building upon the asynchronous TOA algorithm, a hybrid RSS-TOA target location algorithm is developed, resolving different scenarios with unknown transmission parameters. Simulation results indicate its applicability without requiring structural modifications, with enhanced positioning accuracy outweighing the increased computation time. The significance of the research lies in developing innovative algorithms to enhance the positioning accuracy in wireless sensor networks, addressing challenges associated with target and node localization, and offering superior alternatives to available approaches.

Future research will focus on the energy efficiency of the proposed method.

## Funding

This study was supported by the Project of Innovation Fund for Higher Education in Gansu Province in 2022 (No. 2022A-171), Project name: Research and application of ZigBee wireless communication network and MEMS sensor in engineering structure damage monitoring, and by the 2024 Gansu Provincial Talent Project, Project name: Research on Early Warning and Monitoring System of Earthen Ruins Protection in the Yellow River Basin Based on Multi-Source Information Fusion. The funders had no role in study design, data collection and analysis, decision to publish, or preparation of the manuscript.

## Grant Disclosures

The following grant information was disclosed by the authors:
Innovation Fund for Higher Education in Gansu Province in 2022: 2022A-171.

## Competing Interests

The authors declare that they have no competing interests.

## Author Contributions

- Hui Yang conceived and designed the experiments, performed the experiments, analyzed the data, performed the computation work, prepared figures and/or tables, authored or reviewed drafts of the article, and approved the final draft.

## Data Availability

The raw data is available in the Supplemental Files.

## Supplemental Information

Supplemental information for this article can be found online at http://dx.doi.org/10.7717/peerj-cs.2179#supplemental-information.

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
