# Peer review of "Wireless sensor localization based on distance optimization and assistance by mobile anchor nodes: a novel algorithm"

_PeerJ Computer Science, doi:10.7717/peerj-cs.2179_

## Round 0.1 · original submission · Major Revisions

Dear authors,

Thank you for submitting your article. Feedback from the reviewers is now available. It is not recommended that your article be published in its current format. However, we strongly recommend that you address the issues raised by the reviewers, especially those related to readability, experimental design and validity, and resubmit your paper after making the necessary changes. Before submitting the paper following should also be addressed:

1. Please write research gap and the motivation of the study. Evaluate how your study is different from others.
2. Organization of the paper should be given at the end of Introduction section.
3. The values for the parameters of the algorithms selected for comparison are not given.
4. The paper lacks the running environment, including software and hardware. The analysis and configurations of experiments should be presented in detail for reproducibility. It is convenient for other researchers to redo your experiments and this makes your work easy acceptance. A table with parameter settings for experimental results and analysis should be included in order to clearly describe them.
5. Equations should be used with equation number. Please do not use “following”, “as follows”, etc. Explanation of the equations should be checked. All variables should be written in italic as in the equations. Their definitions and boundaries should be explained. Relevant references should be given for the equations.

Best wishes,

Reviewer 1 ·

Basic reporting

localization accuracy in WSNs, making them more suitable for various applications such as healthcare, environmental monitoring, and target tracking.
The abstract needs improvements , it should contains the problem background , proposed solution, the adopted methodology and the results obtained in a clear and concise manner so that the readers could get a clear idea about the innovation and novelty of the paper from the abstract.

The introduction provides a good overview of the significance of WSNs and localization. However, it could be strengthened by including a clearer statement of the specific problem or gap in current localization techniques that the proposed algorithms address.

Clarify how the proposed algorithms/research overcome specific challenges in WSN localization, such as time synchronization and unknown transmission parameters

Just define the abbreviations only once and then use them in rest of the article. As in current manuscript , this is repeated eg for Wireless Sensor Networks (WSNs), it has been repeatedly defined in different sections of the manuscript.

The paper needs to be professionally edited by some experts to better understand the theme of the study.

Address any unexpected outcomes and provide insights into their possible causes.

Discussion:

Interpret the findings in the context of the study objectives and the broader field of WSN localization.

Discuss the implications of the results for practical applications and future research directions.

Highlight the strengths and limitations of the proposed algorithms and suggest possible avenues for improvement.


Conclusion:
Summarize the key findings of the study and their significance.
Reinforce the contributions of the proposed algorithms to the field of WSN localization.

Conclude with a reflection on the potential impact of the research and opportunities for further investigation.

References:
Ensure that all cited works are properly referenced following a consistent citation style (e.g., IEEE, APA).

Include recent and relevant sources to support the discussion and justify the novelty of the proposed algorithms.

By addressing these points, you can enhance the clarity, rigor, and impact of the paper, making it more compelling for readers and researchers in the field of wireless sensor networks and localization.

Expand the discussion section to delve deeper into the implications of the findings.

How do the proposed algorithms contribute to advancing the state-of-the-art in WSN localization? What are the potential real-world applications and practical benefits?

Consider discussing the scalability and robustness of the proposed algorithms, as well as their suitability for deployment in different WSN scenarios

Ensure that all references cited in the paper are listed accurately and formatted correctly according to the chosen citation style (e.g., IEEE, APA).

Experimental design

No comments.

Validity of the findings

No comments.

Additional comments

No comments.

Reviewer 2 ·

Basic reporting

The paper explores localization algorithms for wireless sensor networks (WSNs), crucial for applications such as healthcare, environmental monitoring, and target tracking. The study introduces innovative approaches to enhance the accuracy of both target and node localization within WSNs. This paper could get the wider readership of the audience, however, the following suggestions should be incorporated to further enhance the quality of the manuscript.
The authors must Start with a brief overview of the importance of localization in WSNs, its applications, and the challenges it addresses. Also please Clearly define the scope and objectives of the study.
Provide a roadmap for the paper to guide the reader through the subsequent sections in the introduction section.
The literature review seems to be too short and it needs expansion to provide a comprehensive overview of existing localization algorithms in WSNs.
Discuss the strengths and limitations of previous approaches, highlighting the gap your proposed algorithms aim to address.
Provide a detailed explanation of the proposed asynchronous Time-of-Arrival (TOA) target localization algorithm.
Clearly outline the steps involved, including signal representation, differential evolution algorithm implementation, and optimization criteria.
Explain how the proposed algorithm differs from existing approaches and how it improves accuracy.
Similarly, describe the hybrid RSSTOA target localization algorithm in detail, emphasizing its novel features and how it tackles challenges related to unknown transmission parameters.
Include mathematical formulations and algorithms if applicable to enhance clarity.

Experimental design

Briefly Describe the simulation environment or experimental setup used to evaluate the proposed algorithms.
Specify the performance metrics used for evaluation, such as localization accuracy, computational complexity, and energy efficiency.
Justify the choice of parameters and scenarios considered in the experiments.

Validity of the findings

Present the results of the experiments in a clear and organized manner, using tables, graphs, and figures as appropriate.
Analyze the performance of the proposed algorithms in comparison to existing methods.
Discuss any observed trends, trade-offs, or limitations revealed by the experimental results.

Additional comments

The language of the paper can also be made better.

---

## Round 0.2 · accepted · Accept

Dear authors,

Thank you for clearly addressing all the reviewers' comments. I confirm that the quality of your paper is improved. The paper now seems to be ready for publication in light of this revision.

Best wishes,

Reviewer 1 ·

Basic reporting

The revised version of the paper has no issues.

Experimental design

No comments.

Validity of the findings

No comments.

Reviewer 2 ·

Basic reporting

All the concerns have been addressed. The paper can be accepted in it’s current state.

Experimental design

NA

Validity of the findings

NA

Additional comments

NA